# Potency of an Inactivated Influenza Vaccine against a Challenge with A/Swine/Missouri/A01727926/2015 (H4N6) in Mice for Pandemic Preparedness

**DOI:** 10.3390/vaccines8040768

**Published:** 2020-12-16

**Authors:** Hirotaka Hayashi, Norikazu Isoda, Enkhbold Bazarragchaa, Naoki Nomura, Keita Matsuno, Masatoshi Okamatsu, Hiroshi Kida, Yoshihiro Sakoda

**Affiliations:** 1Laboratory of Microbiology, Faculty of Veterinary Medicine, Hokkaido University, Kita 18 Nishi 9, Kita-Ku, Sapporo 060-0818, Japan; hayashihirotaka@vetmed.hokudai.ac.jp (H.H.); nisoda@vetmed.hokudai.ac.jp (N.I.); bazarragchaa@vetmed.hokudai.ac.jp (E.B.); matsuno@vetmed.hokudai.ac.jp (K.M.); okamatsu@vetmed.hokudai.ac.jp (M.O.); 2International Collaboration Unit, Research Center for Zoonosis Control, Hokkaido University, Sapporo 001-0020, Japan; 3Laboratory of Biologics Development, Research Center for Zoonosis Control, Hokkaido University, Kita 20 Nishi 10, Kita-Ku, Sapporo 001-0020, Japan; nomura@czc.hokudai.ac.jp; 4Research Center for Zoonosis Control, Hokkaido University, Kita 20 Nishi 10, Kita-Ku, Sapporo 001-0020, Japan; kida@vetmed.hokudai.ac.jp

**Keywords:** antigenicity, H4 influenza, immunogenicity, pandemic, vaccine

## Abstract

H4 influenza viruses have been isolated from birds across the world. In recent years, an H4 influenza virus infection has been confirmed in pigs. Pigs play an important role in the transmission of influenza viruses to human hosts. Therefore, it is important to develop a new vaccine in the case of an H4 influenza virus infection in humans, considering that this virus has a different antigenicity from seasonal human influenza viruses. In this study, after selecting vaccine candidate strains based on their antigenic relation to one of the pig isolates, A/swine/Missouri/A01727926/2015 (H4N6) (MO/15), an inactivated whole-particle vaccine was prepared from A/swan/Hokkaido/481102/2017 (H4N6). This vaccine showed high immunogenicity in mice, and the antibody induced by the vaccine showed high cross-reactivity to the MO/15 virus. This vaccine induced sufficient neutralizing antibodies and mitigated the effects of an MO/15 infection in a mouse model. This study is the first to suggest that an inactivated whole-particle vaccine prepared from an influenza virus isolated from wild birds is an effective countermeasure in case of a future influenza pandemic caused by the H4 influenza virus.

## 1. Introduction

Influenza A virus is an enveloped virus containing single-stranded, eight-segmented, negative-sense RNA; it is categorized into 18 hemagglutinin (HA) and 11 neuraminidase (NA) subtypes [1]. HA binds to the glycan of the host cells, but the glycan structure differs among animal species. It is thought that an influenza pandemic may originate from influenza viruses circulating among pigs. Pigs play the role of a “mixing vessel” by expressing both human- and avian-type receptors to influenza viruses in respiratory epithelial cells and cause a reassortment between two types of influenza viruses [2]. Waterfowl are natural hosts of influenza viruses, and viral antigenicity is conserved within the natural host [3]. Therefore, as preparedness against an influenza pandemic, influenza viruses perpetuated in waterfowl, whose antigenicity is highly conserved, have been isolated. Their antigenicity and proliferation have been evaluated since 1977; thereafter, the isolates have been stored (http://virusdb.czc.hokudai.ac.jp). Vaccines against H1, H5, and H9 from our virus library have been previously prepared, and their efficacy in mice and cynomolgus monkeys has been evaluated [4,5,6].

Since their first isolation in Czechoslovakia in 1954, H4 avian influenza viruses have been sporadically isolated worldwide [7]. H4 virus infection in swine has been recently reported. A/swine/Ontario/01911-1/1999 (H4N6) [8] in Canada, A/swine/Hubei/06/2009 (H4N1) [9], A/swine/Guangdong/K4/2011 (H4N8) [10] in China, and A/swine/Missouri/A01727926/2015 (H4N6) (MO/15) in 2015 were isolated [11]. Viruses are classified based on the HA genes. In a previous report, the HA genes of H4 viruses were systematically classified into North American and Eurasian lineages [11]. According to the phylogenetic analysis of the nucleotide sequence of the eight gene segments of the MO/15 virus, all segments were derived from North American birds [11]. H4 viruses, including MO/15 detected in North America, showed binding to α2,6-linked sialylated glycan, which is highly expressed in the respiratory epithelium of swine and humans [11,12]. MO/15 replicated in swine lungs, causing mild lung lesions, but did not replicate in the upper respiratory tract [11]. This may indicate that H4 influenza viruses could be sustained in the pig population by continuous virus infection and might accidentally be transmitted to other species. To date, there have been no human infections with an H4 virus, although a case of human infection with A/Indiana/08/2011 (H3N2)v derived from swine influenza strain was confirmed [13], and H4 viruses could pose a risk of human infection in the future.

Therefore, development of a vaccine against the H4 virus, which has a different antigenicity from conventional seasonal influenza viruses, is required. No report has evaluated vaccines against H4 influenza virus infection. Therefore, this study aimed to select a candidate vaccine strain from our virus library, prepare a vaccine, and evaluate the vaccine’s potency in a mouse model.

## 2. Materials and Methods

### 2.1. Virus and Cells

MO/15 was kindly provided by Dr. Alicia Janas-Martindale (United States Department of Agriculture, Raleigh, NC, USA). A/duck/Czechoslovakia/1956 (H4N6) (Dk/Cz), A/budgerigar/Hokkaido/1/1977 (H4N6) (Budge/Hok), A/duck/Hokkaido/491003/2014 (H4N2) (Dk/Hok/491003), A/duck/Mongolia/769/2015 (H4N6) (Dk/Mon), A/duck/Hokkaido/138/2007 (H4N6) (Dk/Hok/138), A/swan/Hokkaido/481102/2017 (H4N6) (Swan/Hok), and A/mallard/Alberta/223/1979 (H4N2) (Mal/Alb) were isolated from birds. All viruses used in this study were propagated in 10-day-old embryonated chicken eggs at 35 °C for 48 h, and the allantoic fluid was collected and stored at −80 °C until use. Madin–Darby canine kidney (MDCK) cells were maintained in minimum essential medium (MEM; Nissui Pharmaceutical, Tokyo, Japan) supplemented with 10% non-immobilized fetal calf serum (FCS; Sigma-Aldrich, St. Louis, MO, USA), 0.3 mg/mL L-glutamine (Nacalai Tesque, Kyoto, Japan), 100 U/mL penicillin G (Meiji Seika Pharma, Tokyo, Japan), 0.1 mg/mL streptomycin (Meiji Seika Pharma, Tokyo, Japan), and 8 μg/mL gentamicin (Takata Pharmaceutical, Saitama, Japan) in an incubator at 37 °C with 5% CO_2_.

### 2.2. Sequencing and Phylogenetic Analysis

Using TRIzol LS reagent (Life Technologies, Carlsbad, CA, USA), viral RNA was extracted from the allantoic fluid of embryonated chicken eggs and reverse-transcribed with Uni12 primer (5′-AGCAAA AGCAGG-3′) and M-MLV reverse transcriptase (Life Technologies, Carlsbad, CA, USA). The HA genome was amplified by polymerase chain reaction using Ex Taq polymerase (Takara Bio, Shiga, Japan) and a gene-specific primer set. Direct sequencing of HA gene segments was performed using the BigDye Terminator version 3.1 Cycle Sequencing Kit (Life Technologies, Carlsbad, CA, USA) and 3500 Genetic Analyzer (Life Technologies, Carlsbad, CA, USA). Other nucleotide sequence data were acquired from GenBank (https://www.ncbi.nlm.nih.gov/genbank/) and GISAID (https://www.gisaid.org/). Nucleotide sequence data were processed by Genetyx version 15 (Genetyx Corporation, Tokyo, Japan). Phylogenetic analysis of the HA gene was performed by neighbor joining with 1000 bootstrap replications in MEGA 7.0 [14]. The phylogenetic tree of the H4 HA genes was rooted in A/swine/Ontario/01911-1/1999 (H4N6). The genome sequences identified in this study were registered in GenBank/EMBL/DDBJ: A/duck/Hokkaido/W195/2015 (accession no. LC498517), A/duck/Hokkaido/W214/2006 (accession no. LC498518), and A/duck/Hokkaido/W217/2015 (accession no. LC498519). Forty-eight strains of H4 HA genes in this study were clustered into Eurasian and North American lineages based on nucleotide identities. The intragroup homology was between 96% and 99%, whereas the intergroup homology was between 81% and 92%.

### 2.3. Antigenic Analysis

The antigenicity of H4 viruses was evaluated by the hemagglutination-inhibition (HI) test using chicken polyclonal antiserum [15]. Inactivated antigen (500 µg) was immunized to chicken twice at a 2-week interval. After confirmation of more than 2560 HI titers in the serum collected 2 weeks later from the second immunization, 500 µg of antigen was intravenously injected. One week after the last immunization, total blood was collected from the heart, and antisera were prepared. The viruses were diluted to an 8-hemagglutination unit in phosphate-buffered saline (PBS). Then, 25 µL of the diluted virus was added to 25 μL of each antiserum serially two-fold diluted in PBS and incubated at room temperature for 30 min. Then, 50 µL of 0.5% chicken red blood cells in PBS was added and incubated at room temperature for another 30 min. The HI titer was expressed as the reciprocal of the highest serum dilution, showing complete HI.

Based on the HI test results, antigenic cartography was developed using web-based software (http://www.antigenic-cartography.org/) based on the previous research [14,16]. Briefly, each HI titer *N_ij_* was transferred into a table antigenic distance *D_ij_* between virus *i* and antiserum *j* by calculating the difference between the titer for the virus HI by each antiserum *j*, defined as *b_j_*, and the measured titer for each virus *N_ij_* against that antiserum: *D_ij_* = log_2_(*b_j_*) − log_2_(*N_ij_*). To find the map distance, the Euclidean distance *d_ij_* between each virus *i* and antiserum *j* was set, and the difference between the map and table distances was minimized using the error function *E* = ∑*_ij_e* (*D_ij_*, *d_ij_*). The error of a serum-virus pair was defined as *e*(*D_ij_*, *d_ij_*) = (*D_ij_* − *d_ij_*)^2^. The ellipses of the antigenic groups were visually determined, covering all the antigens in each group with the smallest sizes. The spacing between grid lines is equivalent to an antigenic unit distance corresponding to a two-fold HI difference.

### 2.4. Vaccine Preparation

Viruses in allantoic fluid were purified by centrifugation and sedimentation via a sucrose gradient, as described by Kida et al. [17]. The allantoic fluid was ultracentrifuged, and the resulting pellet was layered on 10–50% sucrose density gradient and ultracentrifuged again. Fractions containing the band of virus particles were collected based on sucrose concentration, HA titer, and protein concentration. Whole-virus particles were pelleted from the sucrose fraction by ultracentrifugation and suspended in PBS. The purified virus was inactivated by incubation in 0.1% formalin at 4 °C for 7 days. The inactivation of the formalin-treated virus was confirmed by no virus growth in embryonated chicken eggs after inoculation. The purified inactivated virus was used as a whole-virus particle vaccine. The total protein concentration was measured using the BCA Protein Assay Kit (Thermo Fisher Scientific, Waltham, MA, USA). The amount of vaccine in the potency test was set to 4, 20, and 100 µg based on previous research [6]. According to the ratio between HA protein (14.7 µg) and whole-particle vaccine (50 µg) for influenza A virus [5], the amount of HA proteins in each vaccine was estimated as 1.2, 5.9, and 29.4 µg, respectively.

### 2.5. Pathogenicity of H4 Influenza Viruses in Mice

Eight-week-old female BALB/c mice (Japan SLC, Shizuoka, Japan) were challenged with the virus to evaluate virus pathogenicity in mice. Four mice of four groups were intranasally inoculated with MO/15 at 30 µL/mouse under anesthesia. The mixtures of tiletamine hydrochloride (20 mg/kg) (United States Pharmacopeia, Rockville, MA, USA), zolazepam hydrochloride (20 mg/kg) (United States Pharmacopeia, Rockville, MA, USA), and xylazine (20 mg/kg) (Bayer Yakuhin, Ltd., Osaka, Japan) were injected intraperitoneally into mice within 100 µL [5]. The viral titer of the challenge was 10^5.9^, 10^5.0^, 10^4.0^, or 10^3.0^ plaque-forming units (PFU) of MO/15 (30 µL). The viral titer was originally 10^7.4^ PFU/mL (10^5.9^ PFU/30 µL), and this virus stock was used for the highest-dose group without dilution in PBS. Mice were observed daily for changes in body weight and clinical signs until 9 days post-inoculation (dpi). The humane endpoint was determined as 70% of the body weight at challenge, and if the body weight of a mouse decreased below the endpoint, the mouse was euthanized.

### 2.6. Potency Test of the Vaccine Against MO/15 in Mice

Swan/Hok and MO/15 vaccines with 4, 20, and 100 µg of protein were subcutaneously injected into groups of 10 4-week-old female BALB/c mice (Japan SLC, Shizuoka, Japan). PBS was injected into negative control (NC) and non-vaccinated mice. After 2 weeks, after collecting blood from the tail vein to obtain serum, the same amount of vaccines was injected. At 2 weeks after the second immunization, blood samples were collected from the tail vein, and 10^5.9^ PFU/30 µL of MO/15 was intranasally inoculated to mice under anesthesia. Three days after the challenge, five mice from each group were euthanized, and the lungs were collected for measuring virus recovery. The mice were euthanized by intraperitoneal injection of pentobarbital (150 mg/kg) (Kyoritsu Seiyaku Corporation, Tokyo, Japan). Viral titers in the lung homogenates were measured using a plaque assay in MDCK cells. The other five mice in each group were observed for 14 days for clinical signs.

### 2.7. Serum Neutralization Test

Serum-neutralizing antibody titers in mice were measured using methods described in previous research [6]. Briefly, mouse serum samples were heat-inactivated at 56 °C for 30 min and mixed with 100-fold median tissue culture infectious dose (TCID_50_) of MO/15 or Swan/Hok virus and incubated for 1 h at room temperature. The mixture was inoculated onto confluent monolayers of MDCK cells and incubated at 35 °C for 1 h. Unbound viruses were removed, and cells were washed with PBS. Cells were incubated with MEM containing 5 μg/mL acetylated trypsin (Sigma-Aldrich, St. Louis, MO, USA). Cytopathic effects were observed after 72 h of incubation, and the neutralizing antibody titers were determined as the reciprocal of the serum dilution yielding 50% inhibition of the cytopathic effects.

### 2.8. Virus Titration in Mouse Lungs

Plaque assays were performed as previously described [6]. Briefly, 10-fold dilutions of virus samples or mouse lung homogenates in MEM without FCS were inoculated (100 µL/well) onto confluent monolayers of MDCK cells and incubated at 35 °C in a 5% CO_2_ incubator for 1 h with tilting every 15 min. The virus solution was removed, and cells were washed with PBS once. Cells were then overlaid with MEM containing 5 μg/mL acetylated trypsin (Sigma-Aldrich, St. Louis, MO, USA) and 1% Bacto Agar (Becton, Dickinson and Company, Franklin Lakes, NJ, USA) heated at 42 °C. After 48 h of incubation at 35 °C, cells were stained with 0.005% neutral red with MEM and 1% Bacto Agar. After incubation at 35 °C for an additional 24 h, the number of plaques that were fewer than 100 plaques/well was counted. The number of PFU/g in the original solution was calculated as the product of the reciprocal value of the dilution and the number of plaques in that dilution.

### 2.9. Statistical Analysis

The Student’s *t*-test was used to analyze differences in the body weights of mice, virus recovery, and neutralizing antibody titer between the two groups. One-way analysis of variance was used to analyze differences among multiple groups [18]. Animal survival and body weight data were analyzed using the Mantel–Cox test.

### 2.10. Comparison of Amino Acid Substitutions on the 3D Structure of HA

Amino acid sequence data were acquired from GenBank (https://www.ncbi.nlm.nih.gov/genbank/) and data were processed by Genetyx version 15. X-ray crystallographic structural data of MO/15 (PDB ID: 6V44) [12] was acquired from Protein Data Bank (https://www.rcsb.org/) and visualized by Discovery Studio Visualizer (v20.1.0.19295) (https://discover.3ds.com/discovery-studio-visualizer-download) (Dassault Systèmes BIOVIA, San Diego, CA, USA).

### 2.11. Ethics Statement

All animal experiments were approved by the Institutional Animal Care and Use Committee of the Faculty of Veterinary Medicine, Hokkaido University (18-0035 and 16-0105), and all experiments were conducted per the guidelines of this committee. All applicable international, national, and/or institutional guidelines for the care and use of animals were followed. The Graduate School of Veterinary Medicine, Hokkaido University, has been accredited by the Association for Assessment and Accreditation of Laboratory Animal Care International since 2007.

## 3. Results

### 3.1. Genetic Analysis of H4 Influenza Viruses

The genetic sequence of H4 HA genes was determined and phylogenetically analyzed along with reference sequences available in the database (Figure 1). The H4 virus was roughly divided into North American and Eurasian lineages [19]. The viruses classified into the Eurasian lineage were further classified into Groups 1–4. Groups 1–3 were classified by previous research [19], and Group 4 was defined in the present research. Groups were classified based on the sequence homology of HA genes. Group 1 was composed of isolates from China, Mongolia, and Hokkaido from 2009 to 2014. Isolates from Mongolia and China from 2011 to 2015 were set as Group 2. Group 3 was composed of isolates from China, Mongolia, and Hokkaido from 2006 to 2015. Viruses isolated in Hokkaido in 2017 were set as Group 4 and are systematically close to those isolated in Mongolia and China between 2015 and 2016 with high homology. In the North American lineage, viruses were further classified into several groups [20]. They were classified into Groups 1 and 2, which were set in this study for convenience. A/swine/Ontario/01911-1/1999 (H4N6) and MO/15 were included in Groups 1 and 2, respectively. MO/15 was isolated from pigs but genetically close to the viruses isolated from ducks in North America in 2015.

### 3.2. Antigenic Analysis of H4 Influenza Viruses

Eight H4 viruses from the North American and Eurasian lineages were selected to represent groups based on the phylogenetic tree of the HA gene (Figure 1), and their antigenicity was compared using the cross-HI test (Table 1). Then, based on the cross-HI test results, the antigenic cartography required to project the dataset onto 2D cartography was made (Figure 2). The H4 HA antigenicity was roughly divided into two groups, and all of the classification by the HA gene was not the same as the classification by antigenicity. Swan/Hok, which was isolated in Hokkaido in 2017, and Dk/Cz from the Eurasian lineage or the viruses from the North American lineage were regarded to establish a major antigenic group. Mal/Alb was also included in the major antigenic group. Viruses isolated from budgerigar in 1979 and a virus isolated from ducks in 2007 in Hokkaido were regarded to establish the unique antigenic group. MO/15 was also antigenically close to the major antigenic group. Among them, MO/15 was antigenically closest to Swan/Hok. In all strains, the HA titer of the Swan/Hok inoculum propagated in the allantoic cavity was 512 HA, which indicated that Swan/Hok grew most in embryonated chicken eggs among the viruses used in this study (Appendix A). Therefore, Swan/Hok was selected as the representative vaccine strain to prepare an inactivated whole-particle vaccine.

### 3.3. Pathogenicity of Swine H4 Influenza Virus in Mice

The dose-dependent increase in pathogenicity was evaluated to determine the titer at which clinical signs of mice can be clearly observed (Figure 3). From 2 dpi, compared with the NC group, the body weight of mice in the 10^5.0^ and 10^5.9^ PFU/30 µL groups was significantly reduced. From 3 dpi, the body weight of mice in the 10^3.0^ and 10^4.0^ PFU/30 µL groups was significantly reduced. Compared with the 10^3.0^, 10^4.0^, and 10^5.0^ PFU/30 µL groups, the 10^5.9^ PFU/30 µL group had more severe weight loss, and unlike the other groups, the average body weight decreased to approximately 70% of the body weight at the challenge. Based on these findings, in the potency test of the vaccine, the titer of the challenge virus was decided in sublethal dose of 10^5.9^ PFU/30 µL, making it easy to confirm the effect of clinical signs.

### 3.4. Potency Test of the Vaccine Against H4 Influenza Virus in Mice

Based on the antigenic analysis, Swan/Hok is close to the MO/15 virus and cross-reacts with a wide range of H4 influenza viruses. The titer of the neutralizing antibody in mouse serum vaccinated twice against MO/15 and Swan/Hok was calculated (Table 2). The neutralizing antibody titer of mice immunized with 100 µg Swan/Hok against the MO/15 virus was 1:320. Compared to the 20 µg group, the antibody titers of Swan/Hok were almost 1:160 and slightly higher than the MO/15 group against the MO/15 virus. Compared to the antibody titer of the 20 µg Swan/Hok group, the titer of the serum was similar between the MO/15 virus and Swan/Hok. In these results, the Swan/Hok vaccine showed high immunogenicity and induced the cross-reactive antibody to the MO/15 virus.

Mice vaccinated with each of the two strains were then intranasally inoculated with 10^5.9^ PFU/30 µL MO/15. The average viral titer of the lungs in any vaccination group was significantly lower than in the non-vaccinated groups (Figure 4). The average viral titers of 4 and 100 µg Swan/Hok-vaccinated mice were not significantly different from those with a corresponding amount of MO/15. However, the average viral titer in the lungs of mice vaccinated with 20 µg Swan/Hok was significantly lower than that immunized with the same amount of homologous vaccine MO/15 at 3 dpi. The significant differences between MO/15 4 and 100 µg groups, and MO/15 20 µg and Swan/Hok 100 µg groups were revealed, respectively. In the Swan/Hok group, the average viral titer in the lungs immunized with 20 µg was significantly lower than that with 4 µg at 3 dpi.

Body weight change was observed until 14 dpi (Figure 5). At 3 dpi, a significant difference between all vaccinated groups and the PBS group was observed. There was no significant difference in weight loss at each observation for 14 days between the groups of mice vaccinated with the same amount of MO/15 and Swan/Hok. However, the body weights of mice in the Swan/Hok group recovered earlier than in the MO/15 group; in other words, all Swan/Hok-vaccinated mice gained their body weight to a similar level to those of the NC group, which was earlier than the recovery speed of mice vaccinated with MO/15. The body weights of mice vaccinated with 4 µg Swan/Hok showed no significant difference from those of the NC group after 11 dpi (Figure 5a). At 6 dpi, no significant difference was observed between the body weights of the NC group and those of either 20 or 100 µg Swan/Hok-vaccinated group (Figure 5b,c). Two mice in the non-vaccinated group were euthanized because their body weights reached the humane endpoint, whereas none of the vaccinated mice reached it.

## 4. Discussion

Influenza virus infection in pigs is of great interest due to the potential emergence of a new influenza pandemic strain. For example, the Eurasian avian-like H1N1 swine influenza virus was recently isolated, and its potential ability to infect humans has become a significant concern [21]. Although no human infection with the H4 virus has been confirmed to date, this virus is uniquely positioned to infect humans in the future. In 2015, the H4 virus was isolated from pigs [11]. An influenza pandemic is derived from influenza viruses circulating among pigs. Pigs are infected with both human or avian viruses [22], causing a reassortment between two types of viruses and thereby producing pandemic candidates with an antigenic shift. Because the antigenicity is conserved in waterfowl, which are natural hosts of influenza viruses [23], monitoring the viruses circulating in waterfowl to prepare for an influenza pandemic in humans via pigs is necessary. Therefore, surveillance of waterfowl-derived influenza viruses and vaccines made from influenza virus isolated from birds may be effective in preparing for future outbreaks of a human influenza pandemic. This study is the first report of creating the H4 influenza vaccine and evaluating its potency against the H4 influenza virus.

Based on a phylogenic tree of the HA gene, candidate vaccine strains were screened for a potential pandemic caused by H4 influenza, which has similar virus characteristics to MO/15. A trial vaccine strain was selected based on this phylogenetic tree, antigenic analysis, and proliferation in embryonated chicken eggs. Interestingly, the antigenic analysis resulted in two groups independent of genetic similarity. One of these was the major antigen group composed of Eurasian Dk/Cz, Swan/Hok, and North American Mal/Alb. Although MO/15 was classified in the North American lineage based on the HA gene phylogenetic tree, antigenic analyses placed this virus close to the major antigenic group to which the Eurasian lineage belongs. Regardless of the genetic distance from the North American lineage, the Swan/Hok influenza strain was antigenically closest to MO/15. Furthermore, Swan/Hok showed higher HA titer than other viruses and produced a high final protein concentration in chicken eggs (25.9 µg/egg; MO/15, 14.9 µg/egg). Finally, Swan/Hok was the most proliferative in embryonated chicken eggs among all the candidate strains and, therefore, was the most suitable vaccine candidate.

The vaccine prepared from Swan/Hok induced sufficient neutralizing antibody against MO/15 in mice and reduced the viral load in the lungs. In particular, the neutralizing antibody titer of Swan/Hok-vaccinated mice against a virus challenge was significantly higher than that of the MO/15-vaccinated group at 20 µg. The neutralizing antibody titer of the MO/15-vaccinated group challenged with MO/15 virus was lower in all vaccine concentrations than that of the Swan/Hok group challenged with Swan/Hok virus, demonstrating high immunogenicity of the Swan/Hok vaccine in mice. Furthermore, the neutralizing antibody titer of the Swan/Hok-vaccinated group was similar against homologous and heterologous viruses, indicating that the Swan/Hok vaccine induced a cross-reactive antibody. Between MO/15 and Swan/Hok viruses, there were 20 amino acid substitutions, and 10 amino acid substitutions were located on the head domain (Appendix A). Further analysis is essential to clarify high immunogenicity of Swan/Hok vaccine in mice and cross-reactivity of the induced antibody to MO/15 because limited information is available regarding the structure and antigenic sites of H4 viruses.

MO/15 infection caused an average weight loss of 29% in mice, and a titer of 10^5.4^ PFU/g was measured from the lung homogenate at 3 dpi after a challenge with 10^5.9^ PFU/30 µL. The viral load in the lungs of mice vaccinated with 20 µg Swan/Hok was significantly lower than that of MO/15-vaccinated mice with the same amount. After the virus challenge, no significant difference was observed between the two immunized groups, but the body weight of mice immunized with Swan/Hok recovered earlier than that of the homologously challenged MO/15-vaccinated group. In particular, the body weight of mice vaccinated with 20 µg Swan/Hok recovered at 6 dpi, and this quick recovery was also confirmed in the group receiving 100 µg Swan/Hok. In this study, an inactivated whole-particle vaccine prepared from the Swan/Hok virus selected from our virus library was used in mice after two subcutaneous injections. This vaccine induced sufficient neutralizing antibodies and protected mice against the MO/15 virus challenge. Therefore, the vaccine prepared from the Eurasian avian influenza virus in this study is effective against the H4 influenza pandemic that may occur in the future.

The H4 avian influenza virus was confirmed to be antigenically close to the virus isolated from pigs. The origin of a human influenza pandemic is primarily due to the role of pig as a mixing vessel; therefore, a virus derived from a wild waterfowl that infects pigs has a high potential for causing a future pandemic in humans. It was demonstrated that influenza viruses of both antigenicities are prevalent in waterfowl in Eurasia and North America. The genotypes of H4 viruses were divided into Eurasian or North American lineages, and the antigenicity was not determined by the genotype. A vaccine was prepared, and it produced a significant effect on mice. A virus library of nonpathogenic influenza viruses with 144 combinations of 16 HA and 9 NA subtypes to prepare for influenza pandemic (http://virusdb.czc.hokudai.ac.jp) has been established, and its effectiveness is revealed. Subsequently, an evaluation of a trial vaccine using the strain stocked in the library will be necessary to prepare for an influenza pandemic.

## 5. Conclusions

Swine H4N6 influenza virus MO/15 strain has pandemic potential, considering that it has different antigenicity from seasonal human influenza viruses. The avian-origin H4N6 virus (Swan/Hok strain), which has similar antigenicity to MO/15, was selected from the animal influenza virus library to prepare an inactivated whole-particle vaccine. The prepared vaccine had high immunogenicity in mice, and the induced neutralizing antibodies had high cross-reactivity to the MO/15 virus, resulting in the mitigation of the effects of MO/15 infection in a mouse model.

## Figures and Tables

**Figure 1 vaccines-08-00768-f001:**
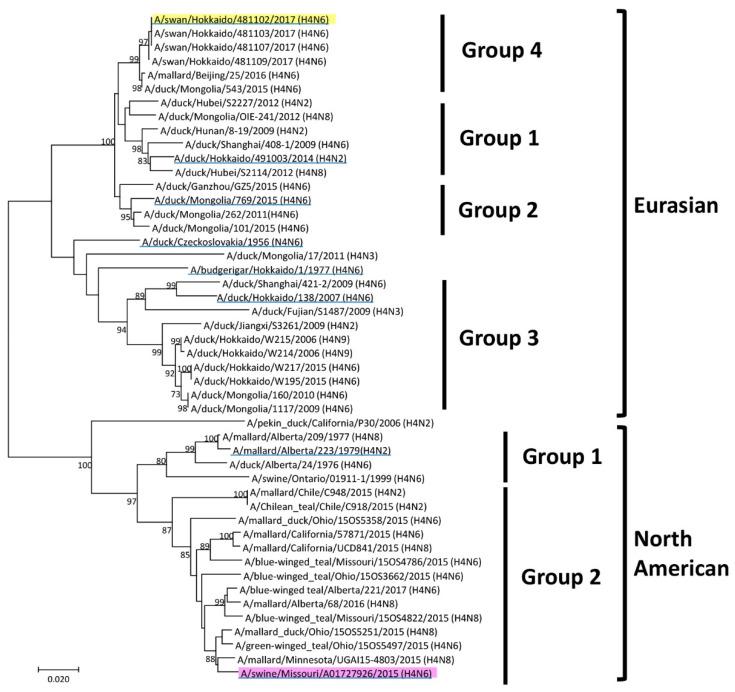
Phylogenetic tree of H4 hemagglutinin (HA) genes of influenza viruses. The nucleotide sequence of the HA gene segment was analyzed by neighbor joining using MEGA 7.0. The number on each node indicates the confidence level of the bootstrap analysis with 1000 replications. The Eurasian lineage was classified into Groups 1–4, and the North American lineage was classified into Groups 1 and 2. The vaccine strain is highlighted in yellow, and the challenging virus is highlighted in pink. The reference viruses used in the antigenic analysis are underlined in blue.

**Figure 2 vaccines-08-00768-f002:**
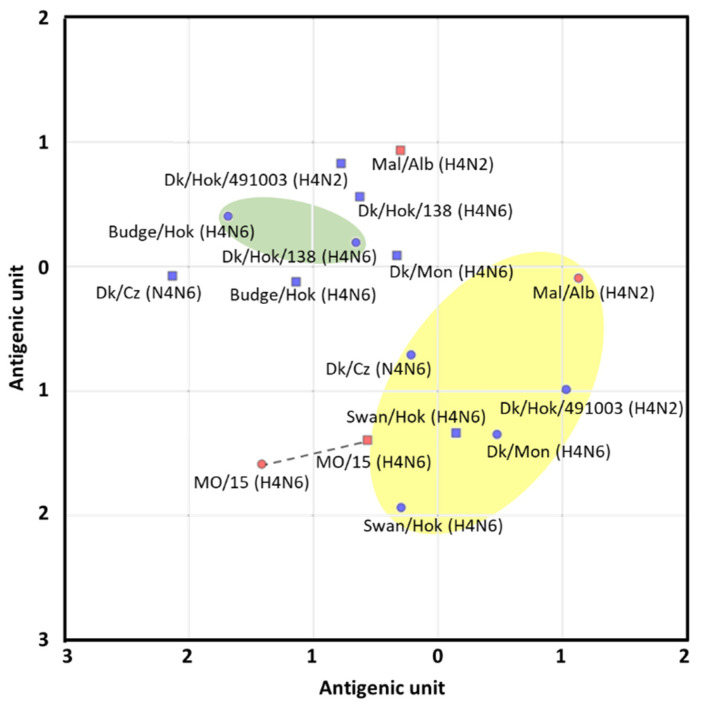
Antigenic cartography based on the cross-HI test of the viruses and antisera of different lineages. In antigenic cartography, vertical and horizontal axes show the distance of the antigen. Dots with round and square shapes indicate antigens and antibodies, respectively. The distance between two dots on the map represents the antigenic distance. Blue and red dots indicate the isolates belonging to the Eurasian and North American lineages, respectively. The antigenic group is shown in yellow or green. The dotted line represents homologous combination. The spacing between grid lines is equivalent to an antigenic unit distance corresponding to a two-fold HI difference.

**Figure 3 vaccines-08-00768-f003:**
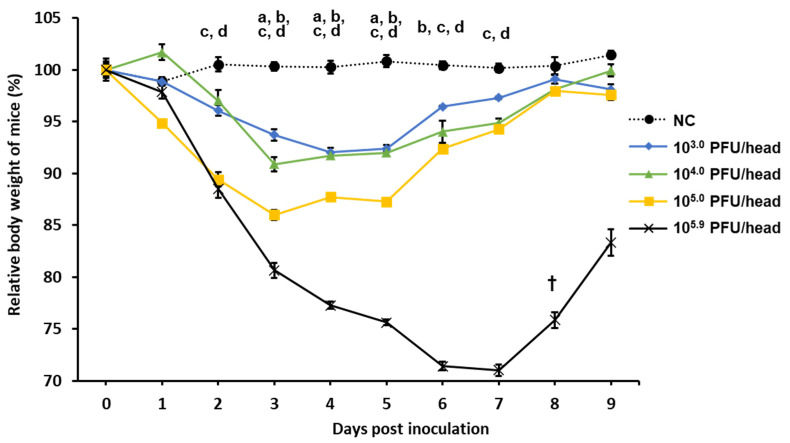
Changes in the body weight of mice inoculated with various doses of MO/15 influenza virus. The pathogenicity of dose-dependent viral titers was compared. Body weight was monitored for 9 days after challenge with different doses (10^3.0^, 10^4.0^, 10^5.0^, or 10^5.9^ plaque-forming units (PFU)/30 µL) of MO/15 (*n* = 4 mice/group). †, euthanasia. a, significant difference between NC and 10^3.0^ PFU/head (*p* < 0.05); b, significant difference between negative control (NC) and 10^4.0^ PFU/head (*p* < 0.05); c, significant difference between NC and 10^5.0^ PFU/head (*p* < 0.05); d, significant difference between NC and 10^5.9^ PFU/head (*p* < 0.05).

**Figure 4 vaccines-08-00768-f004:**
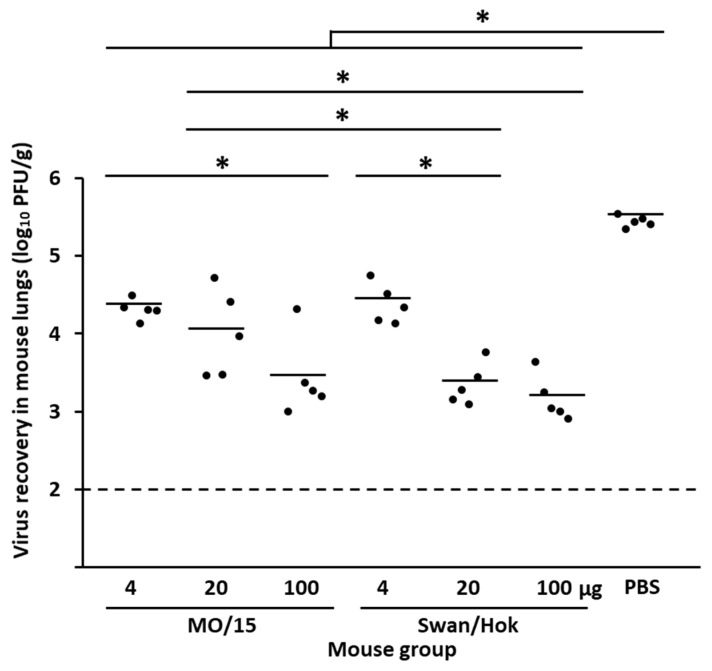
Virus recovery in the lungs of mice subcutaneously vaccinated twice with various amounts of the MO/15 or Swan/Hok vaccine after a challenge with 10^5.9^ PFU/30 µL MO/15. Viral titers were measured on day 3 after the MO/15 challenge. * *p* < 0.05.

**Figure 5 vaccines-08-00768-f005:**
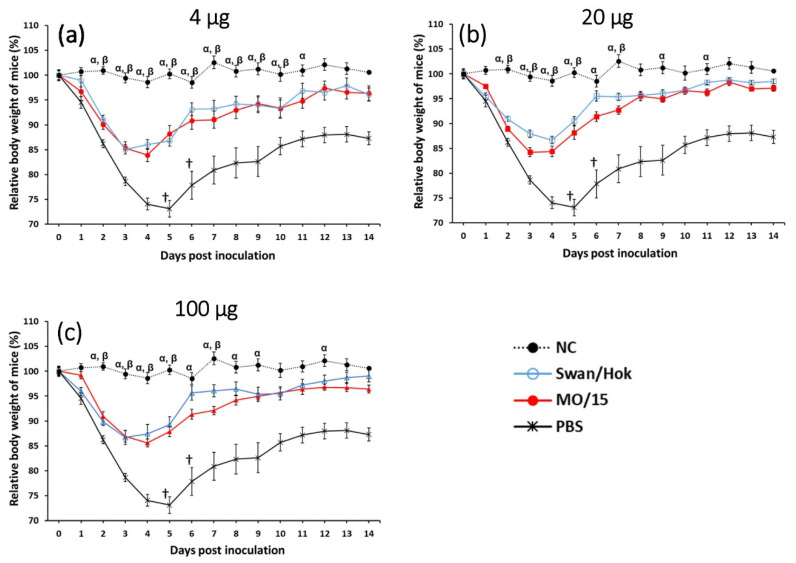
Changes in the body weights of mice subcutaneously vaccinated twice with Swan/Hok and homologous MO/15 after a challenge with MO/15 influenza virus. Body weights were monitored for 14 days after the MO/15 challenge (*n* = 5 mice/group). Mice received two immunizations: 4 µg (**a**), 20 µg (**b**), or 100 µg (**c**) of each vaccine subcutaneously. †, euthanasia. α, significant difference between NC and MO/15 (*p* < 0.05); β, significant difference between NC and Swan/Hok (*p* < 0.05).

**Table 1 vaccines-08-00768-t001:** Cross-reactivity of H4 influenza viruses with chicken antisera to strains in the hemagglutination-inhibition (HI) test.

Group	Viruses	HI Titer of the Antiserum							
		Dk/Cz	Budge/Hok	Dk/Hok/491003	Dk/Mon	Dk/Hok/138	Swan/Hok	Mal/ Alb	MO/15
Eurasian	A/duck/Czeckoslovakia/1956 (H4N6)	5120	10,240	2560	10,240	5120	40,960	1280	10,240
Eurasian	A/budgerigar/Hokkaido/1/1977 (H4N6)	20,480	20,480	2560	5120	5120	2560	5120	5120
Eurasian 1	A/duck/Hokkaido/491003/2014 (H4N2)	1280	2560	560	2560	2560	20,480	1280	10,240
Eurasian 2	A/duck/Mongolia/769/2015 (H4N6)	1280	10,240	320	10,240	1280	40,960	2560	10,240
Eurasian 3	A/duck/Hokkaido/138/2007 (H4N6)	10,240	20,480	5120	10,240	10,240	20,480	10,240	10,240
Eurasian 4	A/swan/Hokkaido/481102/2017 (H4N6)	2560	10,240	640	1280	2560	40,960	640	20,480
North American 1	A/mallard/Alberta/223/1979 (H4N2)	2560	2560	640	2560	2560	20,480	10,240	5120
North American 2	A/swine/Missouri/A01727926/2015 (H4N6)	20,480	5120	640	1280	1280	20,480	2560	20,480

Homologous titers are underlined. Abbreviations: Dk/Cz, A/duck/Czeckoslovakia/1956 (N4N6); Budge/Hok, A/budgerigar/Hokkaido/1/1977 (H4N6); Dk/Hok/491003, A/duck/Hokkaido/491003/2014 (H4N2); Dk/Mon, A/duck/Mongolia/769/2015 (H4N6); Dk/Hok/138, A/duck/Hokkaido/138/2007 (H4N6); Swan/Hok, A/swan/Hokkaido/481102/2017 (H4N6); Mal/Alb, A/mallard/Alberta/223/1979 (H4N2); MO/15, A/swine/Missouri/A01727926/2015 (H4N6).

**Table 2 vaccines-08-00768-t002:** Serum-neutralizing antibody titers of vaccinated mice.

Vaccine	Dose µg	Neutralizing Antibody Titers Against											
		MO/15					GM	Swan/Hok					GM
MO/15	4	80	80	80	80	80	80	40	80	80	80	80	70
	20	80	160	80	160	80	106	40	80	80	80	80	70
	100	320	320	320	320	320	320	80	40	80	80	80	70
Swan/Hok	4	80	80	80	80	80	80	80	80	80	80	80	80
	20	160	160	160	160	160	160	160	160	320	160	160	184
	100	320	320	320	320	320	320	640	160	320	320	320	320
PBS	-	<40	<40	<40	<40	<40	<40	<40	<40	<40	<40	<40	<40

<40: Not detected, GM: geometric mean.

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
