# Peer review of "Potency of an Inactivated Influenza Vaccine against a Challenge with A/Swine/Missouri/A01727926/2015 (H4N6) in Mice for Pandemic Preparedness"

_vaccines, 2020, doi:10.3390/vaccines8040768_

Round 1

Reviewer 1 Report

Hayashi and coworkers study the antigenicity of prototype inactivated whole virus H4 vaccines against a potential zoonotic H4N6 virus from swine with pandemic potential in humans. The study is original and novel and certainly of interest to the field.

Major points:

The method for antigenicity mapping (line 106) should be explained in more detail to make the figure 2 understandable.

It appears that the authors used a sublethal or borderline lethal dose to challenge the mice vaccinated with the two H4 vaccines (Fig. 4). This is quite unusual and raises the question of the potency of these vaccines. For comparison a lethal challenge dose would be appropriate.

Minor points:

Line 59: "It may indicate that H4 influenza viruses should be sustained in 59 pig population[...]" replace 'should' with 'could'

Line 101:

"The eight-hemagglutination unit (25 μL) of the virus was added to 25 μL of a twofold dilution of each antiserum 102 in PBS and incubated at room temperature for 30 min." this sentence needs rephrasing

Author Response

Dear Editor and Reviewers,

Thank you very much for your valuable suggestions and comments for improving the quality of our manuscript in Vaccines. Out manuscript, vaccines-1023395 entitled “Potency of an inactivated influenza vaccine against a challenge with A/swine/Missouri/A01727926/2015 (H4N6) in mice for pandemic preparedness” was revised as follows. Modifications according to Editor are highlighted in green. Modifications according to the reviewer 1’s comments are highlighted in blue. Modifications according to the reviewer 2’s comments are highlighted in yellow. In addition, modifications by ourselves and grammar check company to improve the quality of this manuscript are highlighted in gray.

<Modification of the title>

We excluded the word “the” from our title of the manuscript and changed to “Potency of an inactivated influenza vaccine against a challenge with A/swine/Missouri/A01727926/2015 (H4N6) in mice for pandemic preparedness” under the guidance of the grammar check company.

<Modification of reference papers>

According to the suggestion by Editor and reviewer 2, these previous publications were added in our reference list.

No. 15: World Organisation for Animal Health. Avian influenza (infection with avian influenza viruses). Manual of Diagnostic Tests and Vaccines for Terrestrial Animals. 2018. Avaiable online: https://www.oie.int/standard-setting/terrestrial-manual/access-online/ (accessed on 6 December 2020).

No. 16: Katzelnick, L.C.; Fonville, J.M.; Gromowski, G.D.; Bustos, J.; Green, A.; James, S.L.; Lau, L.; Montoya, M.; Vanblargan, L.A.; Russell, C.A.; et al. Dengue viruses cluster antigenically but not as discrete serotypes. Science 2016, 349, 1338–1343, doi:10.1126/science.aac5017.

No. 23: Bailey, E.; Long, L.; Zhao, N.; Hall, J.S.; Baroch, J.A.; Senter, L.; Cunningham, F.L.; Pharr, G.T.; Hanson, L.; Deliberto, T.J.; et al. Antigenic Characterization of H3 Subtypes of Avian Influenza A Viruses from North America. Avian Dis. 2016, 60, 346–353, doi:10.1637/11086-041015-RegR.

According to the suggestion by Editor, these publications were deleted in our previous reference list.

No. 5: Nishi, T.; Sakoda, Y.; Okamatsu, M.; Kida, H. Potency of an inactivated influenza vaccine prepared from A/duck/Hong Kong/960/1980 (H6N2) against a challenge with A/duck/Vietnam/OIE-0033/2012 (H6N2) in mice. Arch. Virol. 2014, 159, 2567–2574, doi:10.1007/s00705-014-2107-2.

No. 8: Soda, K.; Ozaki, H.; Sakoda, Y.; Isoda, N.; Haraguchi, Y.; Sakabe, S.; Kuboki, N.; Kishida, N.; Takada, A.; Kida, H. Antigenic and genetic analysis of H5 influenza viruses isolated from water birds for the purpose of vaccine use. Arch. Virol. 2008, 153, 2041–2048, doi:10.1007/s00705-008-0226-3.

No. 9: Suzuki, M.; Okamatsu, M.; Fujimoto, Y.; Hiono, T.; Matsuno, K.; Kida, H.; Sakoda, Y. Potency of an inactivated influenza vaccine prepared from A/duck/Mongolia/245/2015 (H10N3) against H10 influenza virus infection in a mouse model. Jpn. J. Vet. Res. 2018, 66, 29–41, doi:10.14943/jjvr.66.1.29.

No. 10: Suzuki, M.; Okamatsu, M.; Hiono, T.; Matsuno, K.; Sakoda, Y. Potency of an inactivated influenza vaccine prepared from A/duck/Hokkaido/162/2013 (H2N1) against a challenge with A/swine/ Missouri/2124514/2006 (H2N3) in mice. J. Vet. Med. Sci. 2017, 79, 1815–1821, doi:10.1292/jvms.17-0312.

No. 19: Le, K.T.; Okamatsu, M.; Nguyen, L.T.; Matsuno, K.; Chu, D.H.; Tien, T.N.; Le, T.T.; Kida, H.; Sakoda, Y. Genetic and antigenic characterization of the first H7N7 low pathogenic avian influenza viruses isolated in Vietnam. Infect. Genet. Evol. 2020, 78, 104117, doi:10.1016/j.meegid.2019.104117.

No. 21: Sakabe, S.; Sakoda, Y.; Haraguchi, Y.; Isoda, N.; Soda, K.; Takakuwa, H.; Saijo, K.; Sawata, A.; Kume, K.; Hagiwara, J.; et al. A vaccine prepared from a non-pathogenic H7N7 virus isolated from natural reservoir conferred protective immunity against the challenge with lethal dose of highly pathogenic avian influenza virus in chickens. Vaccine 2008, 26, 2127–2134, doi:10.1016/j.vaccine.2008.02.001.

No. 22: Hiono, T.; Okamatsu, M.; Yamamoto, N.; Ogasawara, K.; Endo, M.; Kuribayashi, S.; Shichinohe, S.; Motohashi, Y.; Chu, D.H.; Suzuki, M.; et al. Experimental infection of highly and low pathogenic avian influenza viruses to chickens, ducks, tree sparrows, jungle crows, and black rats for the evaluation of their roles in virus transmission. Vet. Microbiol. 2016, 182, 108–115, doi:10.1016/j.vetmic.2015.11.009.

<General comments by Editor>

Comment 1.

Please modify reference as high self-citation.

Answer:

Thank you for your advice. Previously, the self-citation of the reference was 12 in 27. We revisited the references and replaced them with more appropriate ones. Finally, the self-citation of the reference is 5 in 23.

Modification:

According to the suggestion by Editor, this previous publication was added.

No. 23: (Bailey et al., Avian Dis., 2016) at Line 304 on Page 10.

According to the suggestion by Editor, total 7 publications were deleted in our reference list.

Comment 2.

Please add in conclusion.

Answer:

Thank you for your advice. As you pointed out, we have added the conclusions.

Modification:

We rewrote the sentence of Line 357 to 360 on Page 13, and Line 361 to 362 on Page 14.

<Modifications to improve the quality of this manuscript by ourselves>

Regarding Table 1, 2, and Table S1, we changed them to embedded frames in the text at Line 223 on Page 6 and Line 263 on Page 8, since they were pasted with graphic images.

E-mail address of Keita Matsuno (K.M.) was changed to the current version (matsuk@czc.hokudai.ac.jp).

We revised the various sentences highlighted in gray according to the suggestions of grammar check company (https://www.enago.jp/).

Reviewer #1

Comment 1.

The method for antigenicity mapping (line 106) should be explained in more detail to make the figure 2 understandable.

Answer:

Thank you for pointing out. Antigenic mapping in this study was operated based on the previous researches (Smith et al., Science, 2004 and Katzelnick et al., Science, 2016). Methods and formulas were added in the text.

Modification:

We rewrote the text of Line 108 to 114 on Page 3 and Line 115 to 116 on Page 3. Additional information was added at the figure legend of Figure 2, Line 236 to 237 on Page 7.

Comment 2.

It appears that the authors used a sublethal or borderline lethal dose to challenge the mice vaccinated with the two H4 vaccines (Fig. 4). This is quite unusual and raises the question of the potency of these vaccines. For comparison a lethal challenge dose would be appropriate.

Answer:

Thank you for your suggestion. MO/15 was not completely lethal with the inoculation of undiluted allantoic fluid (105.9 PFU/30 µL). This result was shown at Figure 3. Therefore, the undiluted MO/15 was used on the experiment of the vaccine potency test and we could not inoculate viruses with complete lethal dose to mice.

Modification: We rewrote the text of Line 137 to 138 on Page 3.

Comment 3.

Line 59: "It may indicate that H4 influenza viruses should be sustained in pig population[...]" replace 'should' with 'could'

Answer:

Thank you for pointing out. We replaced “should” to “could” as the suggestion.

Modification:

We changed the sentence at Line 57 on Page 2.

Comment 4.

"The eight-hemagglutination unit (25 μL) of the virus was added to 25 μL of a twofold dilution of each antiserum in PBS and incubated at room temperature for 30 min." this sentence needs rephrasing

Answer:

Since the content of one sentence was very long and difficult to understand, it was divided into two sentences. We changed the sentence to “The viruses were diluted to an 8-hemagglutination unit in phosphate-buffered saline (PBS). Then, 25 µL of the diluted virus was added to 25 μL of each antiserum serially two-fold diluted in PBS and incubated at room temperature for 30 min.”.

Modification:

We rewrote the text of Line 101 to 104 on Page 3.

Reviewer #2

Comment 1.

Paragraph 2.2. : How was constructed the pylogenic tree? Are all the sequence produced by the authors? Have they used some reference sequences (GISAID? NCBI?)? How was rooted the tree?

Answer:

A rooted phylogenetic tree was constructed with neighbor-joining (1,000 bootstrap replications). Sequence information of A/duck/Hokkaido/W195/2015 (accession no. LC498517), A/duck/Hokkaido/W214/2006 (accession no. LC498518), and A/duck/Hokkaido/W217/2015 (accession no. LC498519) were registered by ourselves. Other data was quoted from NCBI database and GISAID, and we added the acknowledgements for the citation of GISAID. The phylogenetic tree was rooted to A/swine/Ontario/01911-1/1999 (H4N6).

Modification:

The change was added to the article in Line 85 to 87 on Page 2, Line 89 to 93 on Page 2, and Line 374 on Page 12.

Comment 2.

In case of newly obtained sequences, a SRA accession number has to be indicated.

Answer:

Thank you for your constructive comment. We added the accession numbers of A/duck/Hokkaido/W195/2015 (accession no. LC498517), A/duck/Hokkaido/W214/2006 (accession no. LC498518), and A/duck/Hokkaido/W217/2015 (accession no. LC498519).

Modification:

The sentence was changed at Line 91 to 93 on Page 2.

Comment 3.

In methods, authors have to give more detail about their definition of a group (and/or subgroup) as the subdivision is not that easy to validate. Have the authors based their conclusion on distance ? bootstrap? Sequence homology (if yes which percentage?). Moreover, they have to indicate how many informative sites were considered for analysis in the phylogenetic analysis?

Answer:

Eurasian Groups 1, 2, and 3 were classified according to the previous research (Liang et al., J. Virol., 2016). The grouping was based on nucleotide sequence homology. Virus strains with similar homology and different from Eurasian Groups 1, 2, and 3 were newly defined as Eurasian Group 4. Homology between groups ranged from 81 to 92%, and homology within groups ranged from 96 to 99%. In the previous paper, the homology between groups was 92.2% or less, and the homology within the group was 95% or more, suggesting that these results are almost the same conditions. Informative sites were not shown as they were created by neighbor-joining rather than maximum-parsimony.

Modification:

We changed the sentence at Line 89 on Page 2, Line 94 to 95, and Line 191 on Page 5.

Comment 4.

For part 2.3; 2.5 and 2.6 authors have to give more detail about the protocol they choose to apply. Have all the protocols been published ? How the vaccine dose been chosen?

Answer:

In the chapter 2.3, we added the methodology of HI test (https://www.oie.int/standard-setting/terrestrial-manual/access-online/). The method of creating antigenic cartography was based on the previous researches (Smith et al., Science, 2004 and Katzelnick et al., Science, 2016), and the calculation formula was added in the text. We added the content at Figure 2 that the spacing between grid lines represents a distance of 1 antigenic-unit distance, corresponding to a 2-fold dilution in the HI assay.

In the chapter 2.5, we added a description of the infectivity titer of the inoculum. The criteria to determine the amount of vaccine were added in the chapter 2.4. In addition, since the type, amount, and administration method of anesthesia used in this research were not described, we added to the text in response to the indication (Nomura et al., J. Vet. Med. Sci., 2012). We added the sentences that “Mixture of tiletamine hydrochloride (20 mg/kg) (United States Pharmacopeia, Rockville, MA, U.S.A.), zolazepam hydrochloride (20 mg/kg) (United States Pharmacopeia), and xylazine (20 mg/kg) (Bayer HelthCare, Osaka, Japan) was injected intraperitoneally into mice for anesthesia.”

In the chapter 2.6, as in the chapter 2.5, in addition to the method of administering anesthesia, the method of euthanasia and the drug and method used were added according to the guideline of the Institutional Animal Care and Use Committee of the Faculty of Veterinary Medicine, Hokkaido University. We added the sentences that “The mice were euthanized by intraperitoneal injection of pentobarbital (150 mg/kg) (Kyoritsu Seiyaku Corporation, Tokyo, Japan)”.

The amount of the vaccine was decided based on the previous research (Okamatsu et al., Virol. J., 2013) Amount of HA protein was estimated based on the previous research (Nomura et al., J. Vet. Med. Sci., 2012). The amount of HA proteins in each vaccine was estimated as 1.2, 5.9, and 29.4 µg, respectively since amount of HA protein was generally indicated and discussed to assess the efficacy of vaccine for influenza.

Modification:

Sentences are changed at Line 127 to 129 on Page 3 and Line 133 to 136 on Page 3, Line 149 to 150 on Page 4, and Line 149 to 150 on Page 4.

These previous publications were added in our reference list.

No. 15: (https://www.oie.int/standard-setting/terrestrial-manual/access-online/) at Line 98 on Page 3.

No. 16: (Katzelnick et al., Science, 2016) at Line 108 on Page 3.

Comment 5.

Figure S1 has to be indicated in the main manuscript, as the data it summarized could be as interesting as the other figure.

Answer:

Thank you for pointing out. We moved Figure S1 at main article, and we changed the name as Figure 3. Along with that, the numbers of other Figure numbers in the article have also been changed. The figure legend of Figure 3 was revised for the additional explanations.

Modification:

Figure S1 including the figure legend was moved to the main article, and the name was changed as Figure 3. The number of figures was changed from Figure 3 to Figure 4, and Figure 4 to Figure 5. We added the figure legend of Figure 3, Line 248 to 253 on Page 8.

We deleted incorrect sentences “Taken together, no significant weight loss was confirmed between the groups of non-challenged mice and mice challenged with 105.0 PFU/30 µL or less, but the mice challenged with 105.9 PFU/30 µL only showed significant weight loss compared to the non-challenged mice.” at the chapter 3.3, Line 245 on Page 7.

Comment 6.

Figure 3. Has difference between MO/15 4 and 100 been tested? Similar, has the difference between MO/15 20 and Swan/hok 100 been tested? It is surprising that these values were not statistically different.

Answer:

Significant difference between MO/15 4 and 100 µg groups was revealed. In addition, significant difference between MO/15 20 µg and Swan/Hok 100 µg group was also revealed. Therefore, we added these results in the text and modified Figure 4 to indicate significant differences.

Modification:

We added the sentence at Line 270 to 272 on Page 9. Figure 4 was modified to the revised manuscript.

Comment 7.

A supplementary analysis/figure could be added to the manuscript, to try to explain the observed difference of immunization between strains. Could the authors produce a figure to find antigenic site that have vary between strain and could explain this observation? 

Answer:

In response to the indication, we made a diagram showing the differences in amino acids in the 3D structure as below. The amino acid identity of MO/15 and Swan/Hok HA was 94%, and the similarity of the amino acid was 99%. When comparing the amino acid differences, there were a total of 20 different amino acid residues, and there were 10 amino acid differences in the head domain. The amino acid difference at the head domain was highlighted in yellow, and the amino acid difference at the stalk domain was highlighted in green. However, there was no previous report to identify the antigenic site of H4 virus. Hence, it seems difficult to discuss the high immunogenicity of Swan/Hok vaccine in mice and the cross-reactivity of the induced antibody, suggesting additional researches are needed in the future. Taken together, we did not add this 3D structure in this paper, but the difference of the amino acid residues and the limitation of this study has been added to the discussion part.

Modification:

We add the sentence about the difference of the amino acid residue at the discussion, Line 328 to 332, Page 11.

Reviewer 2 Report

This article By Hirotaka Hayashi et al. described very interesting data on the performances of a newly developed H4N6 vaccine, to prepare to a potential zoonotic pandemic.

Some revision is needed before publication.

Methods:

Paragraph 2.2. : How was constructed the pylogenic tree? Are all the sequence produced by the authors? Have they used some reference sequences (GISAID? NCBI?)? How was rooted the tree?

In case of newly obtained sequences, a SRA accession number has to be indicated.

In methods, authors have to give more detail about their definition of a group (and/or subgroup) as the subdivision is not that easy to validate. Have the authors based their conclusion on distance ? bootstrap? Sequence homology (if yes which percentage?). Moreover, they have to indicate how many informative sites were considered for analysis in the phylogenetic analysis?

For part 2.3; 2.5 and 2.6 authors have to give more detail about the protocol they choose to apply. Have all the protocols been published ? How the vaccine dose been chosen?

Figure S1 has to be indicated in the main manuscript, as the data it summarized could be as interesting as the other figure.

Figure 3. Has difference between MO/15 4 and 100 been tested? Similar, has the difference between MO/15 20 and Swan/hok 100 been tested? It is surprising that these values were not statistically different.

A supplementary analysis/figure could be added to the manuscript, to try to explain the observed difference of immunization between strains. Could the authors produce a figure to find antigenic site that have vary between strain and could explain this observation? 

Author Response

(The authors gave the same response as above.)

Round 2

Reviewer 1 Report

Since lethal challenge is not possible due to the titers of the virus stocks I propose to specifically state in that manuscript that a sublethal challenge was performed.

Otherwise I thank the authors for the corrections. Very nice study.

Author Response

Dear Editor and Reviewers,

Thank you very much for your valuable suggestions and comments for improving the quality of our manuscript in Vaccines. Out manuscript, vaccines-1023395 entitled “Potency of an inactivated influenza vaccine against a challenge with A/swine/Missouri/A01727926/2015 (H4N6) in mice for pandemic preparedness” was revised as follows. Modifications according to the Reviewer-1’s comment is highlighted in blue. Modifications according to the Reviewer-2’s comments are highlighted in yellow.

Reviewer #1

Comment 1.

Since lethal challenge is not possible due to the titers of the virus stocks I propose to specifically state in that manuscript that a sublethal challenge was performed.

Answer:

Thank you for pointing out. We added the words “sublethal dose” at the result 3.3.

Modification:

We added the sentence at Line 251 on Page 7.

Reviewer #2

Comment 1.

The manuscript became suitable for publication but one of my previous comment has not be completely adressed.

Even if the number of amino-acid substitution (20 and 10) has been clearly indicated, a representation of the region of the full protein has to be added. Even if H4 viruses are not completely understood (and limited information is available) a figure could be useful to understand impact of these substitution, by analogy to other influenza viruses. Please add such a figure.

Answer:

Thank you for pointing out. We added the figure of the amino acid substitutions on the HA between MO/15 and Swan/Hok in supplementary file. According to adding extra figure, we added the sentence for making 3D structure and comparing the amino acid substitutions at the chapter of materials and methods.

Modification:

The figure of amino acid substitutions on the HA was added at the supplementary file as Figure S1. We add the sentences at Line 178 to 182 on Page 4. The quotation of Figure S1 was added at Line 336 on Page 11. As the adding new method, the number of subchapter was changed.

Reviewer 2 Report

The authors have applied extensive revisions of their manuscript.

The manuscript became suitable for publication but one of my previous comment has not be completely adressed.

Even if the number of amino-acid substitution (20 and 10) has been clearly indicated, a representation of the region of the full protein has to be added.Even if H4 viruses are not completely understood (and limited information is available) a figure could be useful to understand impact of these substitution, by analogy to other influenza viruses. Please add such a figure.

Author Response

(The authors gave the same response as above.)
